# A Concise Review of the Components and Properties of Wood–Plastic Composites

**DOI:** 10.3390/polym16111556

**Published:** 2024-05-31

**Authors:** Zuzana Mitaľová, Dušan Mitaľ, Khrystyna Berladir

**Affiliations:** 1Department of Automobile and Manufacturing Technologies, Faculty of Manufacturing Technologies, Technical University of Košice, Bayerova 1, 08001 Prešov, Slovakia; dusan.mital@tuke.sk (D.M.); khrystyna.berladir@tuke.sk (K.B.); 2Department of Applied Materials Science and Technology of Constructional Materials, Sumy State University, Rymskogo-Korsakova 2, 40007 Sumy, Ukraine

**Keywords:** natural fibers, wood, wood–plastic composite

## Abstract

This article summarizes findings in the field of the history, composition, and mechanical properties of WPCs (wood–plastic composites) formed by combining two homogeneous substances, i.e., a polymer matrix with cellulose fibers in a certain ratio (with the addition of additives). In relation to a wide range of applied natural reinforcements in composites, it focuses on wood as a fundamental representative of lignocellulosic fibers. It elucidates the concept of wood flour, the criteria for its selection, methods of storage, morphological characteristics, and similar aspects. The presence of wood in the plastic matrix reduces the material cost while increasing the stiffness. Matrix selection is influenced by the processing temperature (*T_max_* = 200 °C) due to the susceptibility of cellulose fibers to thermal degradation. Thermoplastics and selected biodegradable polymers can be applied as matrices. The article also includes information on applied additives such as coupling agents, lubricants, biocides, UV stabilizers, pigments, etc., and the mechanical/utility properties of WPC materials. The most common application of WPCs is in automotive sector, construction, aerospace, and structural applications. The potential biodegradability and lower cost of applications featuring composite materials with natural reinforcements motivated us to delve into this type of work.

## 1. Introduction

Wood–plastic composite (abbreviated as WPC) material comes into existence through a combination of polymers, fillers, and modifiers that can affect the final properties of the composite based on the manufacturer’s requirements (Figure 1). Interest in these composite materials is growing in terms of both industrial applications and basic research. Wood–plastic material offers a few advantages: good insulation, thermal and acoustic properties, manufacturability at relatively low input costs (compared to the production of petroleum-based fiber composite materials)/energy inputs, biodegradability, and healthier working conditions of production. The main disadvantages of WPC products are [1,2]:The significant incompatibility of the polymeric matrix and the natural fillers, which leads to uneven dispersion and reduced mechanical properties values (by adding suitable additives, the properties can be favorably modified);Changes in color, which, unlike natural wood, settles within a few days;Higher acquisition costs (the ratio of initial investment and the service life of the product is ultimately favorable).

A filler in the form of wood flour (of varying faction = particle diameter) or short/long cellulose fibers of plants are applied in WPC mixtures. WPC material allows the use of raw waste wood materials, regardless of their quality. A significant limitation of WPC processing is the processing temperature pertaining to the thermal stability of wood (approximately 200–220 °C), while it is true that with the increasing content of the wood component, this temperature decreases. Up to 97% of manufactured products use the production technology of extrusion. Of all the plastics, polyethylene, polypropylene, polyvinyl chloride, and acrylonitrile-butadiene-styrene copolymers are the most widely applied to matrices. 

Significant milestones in the area include the following [3,4,5]:In 1906, wood flour was used for the first time as an additional component in adhesives. The first composite with an organic filler and a phenol-formaldehyde matrix, referred to as bakelite (used on the knob in a Rolls Royce car);Since 1970, the Italian company GOR Applicazioni Speciali S. p. A. began to use wood fillers and resin composite material (in a ratio of 50/50) in car interior panels;Similarly, Italian extrusion in the production of material (car components) consisting of a polypropylene matrix and an organic filler (50% wood flour) occurred;In 1991, the first conference on organic-filled plastics was held (Madison, USA) with the participation of 50 research workers and the manufacturers themselves. Then, 12 years later, the 7th International Conference on WPCs was held in the same city with 400 participants in attendance;In 1993, the Andersen Corporation (Bayport, MN, USA) began producing wood-filled plastics with a PVC matrix used for the production of door components (the components contained 40% wood reinforcement); three years later, a US company grouping participated in the development of equipment for pallet production;In the 1990s, the WPC market expanded; the material started to be used for flooring in industrial zones, picnic tables, door frames, and beams. In the same period, the Strandex Corporation (Madison, WI, USA) patented a technology for extruding profiles with a high content of wood fibers (about 70% in volume) without the need for finishing technologies;The beginning of the 21st century marked an increase in the demand for these composite materials, and their production increased by 14% annually in the EU and 18% in North America. Over the last 5 years, there has been a 50% increase in interest in WPCs in the area of construction. Over the same time horizon, there has been a 15% increase in interest from automotive manufacturers in materials with natural reinforcement.

Currently, WPC material is beginning to replace rare wood species (Teak, Cumaru, and IPE) in the area of flooring, exterior tiles, and garden furniture. On the other hand, manufacturers themselves also apply these composites in the shipbuilding/automotive industry (dashboards, armrests, seat parts, car cabin trims, etc.). Currently, automotive transportation is responsible for a significant portion of greenhouse gas emissions (CO_2_: carbon dioxide, CH_4_: methane, and NO_x_: nitrogen oxides), contributing to global warming. In an effort to comply with the regulations and directives from the EU, manufacturers are seeking to reduce the weight of vehicles to lower CO_2_ emissions. Reducing vehicle weights by 100 kg leads to a decrease in fuel consumption and a reduction of carbon dioxide emissions by 6 g/km traveled. Reductions in vehicle weights can be achieved through the application of composite materials with natural reinforcements (based on WPCs). Constantly increasing environmental pressures, new ways of using composite materials on a natural basis, as well as technical innovations lead manufacturers to the application of those composites, with the goal of reducing the consumption of financially demanding, non-recyclable types of reinforcements e.g., fiberglass [3,6,7].

## 2. Components of Composite Materials Based on WPCs

### 2.1. Organic Fillers

Natural fibers are divided into categories based on their origin: lignocellulosic materials (wood/non-wood/plant fibers), animals, or minerals. Only plant cellulose fibers obtained from various parts of plants (the seed, leaf, fruit, bast or stem, and stalk) are used as reinforcements for natural fiber-reinforced plastic (NFRP) materials. They can be used in the form of crushed grains and short and long fibers. Natural fibers (NFs) have a low density and weight, do not cause significant tool wear during machining, are non-toxic/biodegradable and their price per unit of volume is lower compared to glass fibers. However, their specific stiffness and strength do not affect that of synthetic fibers. A significant disadvantage of NFs is hydrophilicity, moisture absorption, insufficient polymer wettability (resulting in the incompatibility of components), a high FSI index, as well as limited choices of matrices due to a risk of fiber decomposition under high temperatures during processing. For the classification of natural fibers, see Figure 2 [8,9,10,11].

### 2.2. Wooden Fillers

In regards to wooden reinforcement, wood (>10,000 tree species) can be understood as a specific representative of lignocellulosic fibers. In terms of chemical composition, wood is a hygroscopic material, i.e., a material that can adsorb and/or desorb moisture to the surrounding environment. Other lignocellulosic materials i.e., plant fibers, are similarly hygroscopic. They differ by the degree of absorbency. However, the basic hygroscopicity concept is identical. For this reason, the next part of this paper will operate with wood as the basic representative of lignocellulosic fibers. In the commercial processing of WPC materials, wood flours of various types of wood (soft and hardwood), plant fibers (cotton, loofah, sisal, flax, hemp, kenaf, ramie, coconut, bamboo, palm leaves, banana, rice, corn and barley peels, etc.), and various types of starches are applied [13,14,15].

Many scientific studies have pointed to the possibility of using waste from the agricultural industry as an alternative to wood flour, achieving similar properties to WPC products (Bledzki et al., 2010; Kumar et al., 2010; Hornsby et al., 1997). Similarly, recycled newsprint paper with the addition of modifiers can be used as a source of cellulose fibers [16,17,18].

The term wood flour refers to crushed wooden grains, the texture of which is similar to ordinary food flour. The density (specific gravity) of wood flour is about 1.3–1.4 g·cm^−3^. The particles are sorted by size; they pass through sieves with different mesh diameters, and then they are sorted according to the applicable US standard. For example, the standard assigns a particle that is 850 μm in diameter a mesh 20. The mesh-to-particle diameter converter is shown in Table 1 (the international sieve chart). Commercially, wood flour has been produced since 1906; in the automotive industry, it was first applied in the pre-war period to the production of a material called bakelite (a composite material with a formaldehyde matrix and wood reinforcement). Published studies have reported that the optimal particle size of wood flour is 60/80 mesh (the particle size varies depending on the application). An increase in the particle size of wood flour typically provides a better flow of molten blends and higher flexural moduli. In American states, wood flour for the production of composite materials with a thermoplastic matrix is usually obtained from trees such as pine, oak, and maple (red maple flour increases the water resistance of the resulting material for applications in extremely humid environments and Guayele resin can be applied to increase the resistance to insects/microorganisms). The technology of commercial wood flour production consists of particle size reduction (e.g., using a hammer mill) and subsequent particle classification in vibrating/rotating/oscillating sieves. In fiber production, a drying phase is also required. The common H_2_O content in the fiber is 5–15%; high humidity reduces mechanical properties and thermal stability and induces the biodegradation of the final composite. For wood flour, it is necessary to keep the humidity below 1%. In relation to this, it is also necessary to adapt its storage in closed plastic bags until it is mixed with the polymer [1,2,4,19,20,21].

The final properties of WPC products are influenced by the morphological characteristics of the particles (the shape, particle diameter, length/diameter ratio) and the type of wood from which the flour is made. A low aspect ratio facilitates easy transport to the press barrel but reduces the strength properties of the WPC material. A study on the influence of particle diameter on the resulting properties of composite material was carried out by Salemane and Luyt, when applying flour with a larger fraction, the values of the mechanical properties were measured, and tensile/bending strength was higher in the composite material) [22,23].

**Table 1 polymers-16-01556-t001:** Conversion table for the mesh versus diameter of particles [24].

**US Standard Mesh**	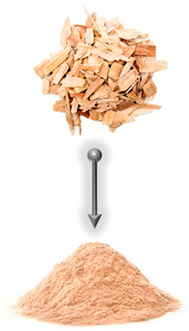	**Diameter of Particle**
20		841 µm
30		595 µm
40		420 µm
50		297 µm
60		250 µm
70		210 µm
80		177 µm
100		149 µm
120		125 µm
140		105 µm
170		88 µm
200		74 µm
230		63 µm
270		53 µm
325		45 µm
400		37 µm

This was followed by more detailed studies dealing with the technological process of wood flour production and its impact on the final nature of the product (wet/dry crushing in a ball mill at different time intervals, the subsequent drying of particles using heat or frost). Detailed SEM images of wood flour particles are included in the study by K. Murayama et al., 2019. Obvious differences in particle morphology are visible only at longer crushing intervals (120 min). Similarly, the change in geometric characteristics is noted with a different method of drying (at a crushing time of 120 min). At shorter intervals, the particles are observed without significant changes. From the point of view of the mechanical properties, the conclusions are as follows [21,22,23]:The values of the notch toughness of the WPC material increase with increasing time intervals of the wood flour crushing process;The highest values of the tensile strength and bending strength of WPC products were recorded when applying particles crushed for 30 min (regardless of the method of the particles’ drying).

### 2.3. Polymeric Matrices

The basic functions of the matrix are to transfer stress to the reinforcement, keep the fiber in the desired position, protect the reinforcement from external influences (moisture, fungi/insect attack, etc.), and increase the resistance of the composite to mechanical degradation caused by abrasion. In the case of WPC materials, the choice of matrix depends on the processing temperature, as organic materials are susceptible to thermal degradation (they resist temperatures up to 200–220 °C). Of the available polymers, the following are suitable: various forms of polyethylene (PE, LDPE, HDPE, etc.), polypropylene (PP), polyvinyl chloride (PVC), and polystyrene (PS). For thermosets (phenolic, epoxy, and polyester resins), see Figure 3.

Studies are currently underway to verify the use of polymethyl methacrylate (PMMA) and Nylon 6. In the case of the above-mentioned matrices, the limiting factor is their biodegradability. Biodegradable polymers from renewable sources (derived from vegetable sources or waste) can be used as an adequate substitute for thermoplastics. This group includes polylactic acid (PLA)-based polymers, polyhydroxyalkanoates (PHA, PHB, and PHBV), starch derivatives, cellulose, esters, and others. Wood fibers or recycled cellulose fibers show good compatibility with PHBV. In conjunction with the PLA matrix, it is possible to apply jute, abaca, flax, wood, hemp, kenaf, and bamboo fibers (A PLA matrix with 40% hemp fiber has a tensile strength up to 87 MPa, Young’s modulus of 7440 MPa, and an impact strength of 20 kJ·m^−2^. The material shows great potential and is suitable for construction applications). A major disadvantage of PLA matrix composite materials is the low thermal stability and toughness [2,26,27,28,29]. 

### 2.4. Additives

The main functions of additives are to improve the mechanical properties of the final product, provide chemical stability, and simplify the process of mixing the components (the improvement of rheological properties), which is illustrated in Figure 4. It is difficult to describe the wide range of additives used in the production of WPCs in view of their diverse resulting properties, method of production technology, and final application. Several types of additives that are most frequently applied were selected from the available studies. Coupling agents, also called compatibilizers, are used to ensure the sufficient dispersion of the components of the mixture and improve their mutual adhesion. Their application results in the linking of two phases i.e., the hydrophilic wood and the hydrophobic polymer phase. They also help to improve the dimensional stability and impact resistance and reduce the risk of fracture under long-term loads [2,4,30]. 

Sometimes they are defined as “wetting wood fiber” agents. Based on the documented research, more than 40 types of coupling agents are currently applied in WPC production. The most used are anhydrides, isocyanates, silanes, and anhydride-modified copolymers (e.g., MAPP: maleic anhydride polypropylene, SEBS-g-MAL: maleinated styrene-ethylene/butylene-styrene, SMA: styrene-maleic anhydride, etc.). The choice of coupling agent depends on the type of filler and the matrix. In the application of coupling agents, it is possible to proceed as follows [7]:(a)Coupling agents are applied to the composite material components in the mixing phase; the process is called a one-step process;(b)Coupling agents are applied to the wood fibers before the components are mixed (two-step process);(c)Coupling agents are applied to a certain volume of wood fibers and polymer granulate as a concentration batch (the resulting mixture is formed by diluting the concentrate).

The first publication dealing with the application of coupling agents to WPC boards was a paper titled “Crosslinking affects the sanding properties of wood–plastics” (Meyer, 1968). A series of patents followed (Gaylord, 1972 applied MAPP in a PP/PVC matrix with a cellulose reinforcement; Coran and Patel, 1982; Geottler 1983) and the application of the first silane agent (Xanthos, 1983) [31,32,33,34,35].

**Figure 4 polymers-16-01556-f004:**
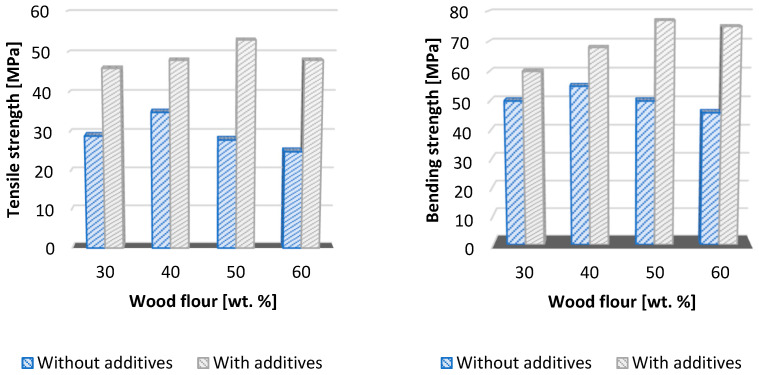
Change in the mechanical properties (tensile and bending strength) after the application of the additive (SCONA series, BYK Additives and Instruments) [36].

Lubricants (rheology control additives) are chemical substances affecting the rheological properties of mixtures, which is the surface quality of the extruded profiles. To some extent, they affect the service life of technological devices, including molds and extruder screws. Based on their function, they can be divided into two groups [30]: External lubricants, which help to move the mixture in the melting screw barrel and ensure that it does not stick to the functional parts of the press (the molecules of the external lubricants are incompatible and do not mix with the mixture), as well as reduce the temperature of the process. These include paraffins, zinc stearates, and PE waxes;Internal lubricants, which reduce the shear forces and viscosity of the thermoplastic matrix at high shear rates and improve the melt viscosity. Acid esters and alcohols are included here; they are compatible with the mixture at high temperatures.

When the organic component of the composite is attacked by a wood-decaying fungus-causing rot, the weight loss of wood in composite material is significantly smaller compared to solid wood. The wood particle is partially protected by the plastic matrix (the particle is encapsulated in plastic). Biocides/biostabilizers are added to ensure the stability of materials against biological attack. Organic (more expensive, with a higher cost per unit weight of the mixture) and inorganic products (e.g., zinc borate, borax, and disodium octaborate tetrahydrate) are available on the market. In addition to the common properties of WPC product additives, biocides must meet the requirements of: Efficacy against a wide spectrum of microbial activity (fungi, mold, and bacteria);Efficacy at low concentrations;Harmlessness to higher organisms;Safety used.

Studies have clearly shown that with increasing wood content, the susceptibility of the composite to fungal mold increases, and thus the application of pesticides is justified [37,38].

Pigments have an aesthetic role; they complement the character of the surface and create a feeling that it is classic wood. They can also play the role of light stabilizers, helping to protect against UV degradation and adverse weather conditions. The classifications of colorants include organic and inorganic pigments as well as dyes [39,40].

Flame retardants are used to reduce the risk of product burning, as both the main components of WPCs are flammable. Exterior retardants must not be susceptible to decay by the weather and cannot be combined with some additives such as HALSs. They are added to the raw material before the process itself. The stabilizers include antioxidants that prevent oxidative reactions, heat stabilizers (phenols), and light stabilizers (HALSs: hindered amine light stabilizers) [38,40].

Density reduction additives, including foaming agents (some available sources also work with the expression of blowers), reduce the weight and costs of the production process, increase the impact strength, and ensure the shape accuracy of profiles, e.g., the sharpness of the corners and contours. When applying exothermic foaming agents, the surface quality was improved, and the WPC product density was reduced by about 30% (when applying foaming agents, it is necessary to use a tandem extruder). The designation of the chemical foaming agents are as follows: CBAs: chemical blowing agents and PBAs: physical blowing agents. The density of WPC boards without the addition of foaming agents is about a third higher than the density of solid wood (WPC boards with PVC matrices: *ρ* = 1.03 g·m^−3^, with the HDPE matrix: *ρ* = 0.95 g·cm^−3^, and with the PP matrix: *ρ* = 0.9–1.05 g·cm^−3^; these values are influenced by the type of wood reinforcement) [37,41,42,43].

Fillers are used to reduce the price of the material, increase the output speed of the production process, and improve the stiffness, durability, and water resistance. On the other hand, they adversely increase the weight of the final profile and reduce the impact resistance. The most common fillers available are talc (it is necessary to consider its particle morphology in its application; some particle shapes can significantly weaken the properties of the final product e.g., an amorphous shape), glass fibers, calcium carbonate, and clay. Recommended percentages of individual additives of WPC material with the PE matrix are shown in Table 2 [2]. 

The final characteristics of WPC products depend on several attributes: the type of polymer matrix applied, the percentage of organic reinforcement (wood and plant fibers), the morphology of the particles themselves, their physical properties and moisture content, the number of individual additives, the technology and conditions of the production process, the origin of the raw input material (the possibility of recycled plastic material applications), the site of natural fiber collection, etc. 

## 3. Properties of Wood Plastic Composites

For manufacturers, the following properties are critical: flexural strength (MOR), flexural modulus (MOE), tensile strength, and shear strength. From the consumer’s point of view, durability, water and fungal resistance, and slip resistance are important [2,4].

The critical factor in WPC design is the ability to create a bond at the components’ wood i.e., the plastic interface. To ensure mutual compatibility, coupling agents that have a significant impact on a number of properties are applied. In the initial studies dealing with changes in mechanical properties in relation to the application of a coupling agent, the MAPP additive was applied to the WPC mixtures to be formulated. Clemons and Caufield, 2005 pointed to a change in the tensile strength and flexural strength after the application of this coupling agent. The increase in strength was 1.5 to two-fold in some cases. The recommended MAPP amount varies depending on the source documents, the type of plastic matrix, and the type of wood reinforcement. Later, the additive portfolio expanded, as evidenced by documented publications. An experimental study by Matheus Poletto, 2017 investigated the effects of two coupling agents, PPgMA (molecular weight: 42 g·mol^−1^) and PPgIA (molecular weight: 90 g·mol^−1^), in combination with recycled polypropylene and wood flour from Pinus elliotti in a study abbreviated as PIE—see Table 3. To improve WPC product compatibility, Rao et al., 2018 applied three types of coupling agents (MAPE, Si69, and VTMS), which significantly helped to encapsulate the wood particles with plastic and fill the cell cores. The dissertation of Saira Taj, 2010 pointed to the dependence of tensile strength on the matrix used and the content of the organic component [20,44,45,46]. 

Similarly, it is possible to increase the flexural modulus by applying additives in small volumes; this is documented in the studies of the additive manufacturers themselves (e.g., Sanyo Chemicals, a substance with the technical name of UMEX). A reduction in the flexural strength and flexural modulus also occurs by reducing the specific gravity of WPCs. The modulus of tensile strength depends on the geometry and fiber ratio of the composite material. In the case of fiber-filled polymers, the tensile modulus is relatively insensitive to the degree of interaction between the fiber and the polymer (Stark and Rowlands, 2003). The shear strength and impact strength are similarly dependent on the matrix material used and the amount/type of the organic component or geometry of the applied fibers [1,22,47].

In general, WPC materials are considered waterproof (not susceptible to biological degradation). The plastic matrix encapsulates the particles and slows down the process of moisture absorption. However, when installing materials outdoors, their susceptibility to moisture, wood-decaying fungi, mold, and insects increases (Mankowski and Morell, 2000). If the moisture content of the wood flour of the composite material exceeds the saturation point (the fiber saturation point), which is approximately 30%, the reinforcement particles are attacked by mold (Xanthos, 2010). The dimensional stability and mechanical properties of WPC materials depend on the moisture content (in the case of insufficient compatibility between the matrix and the filler, microcracks arise, which spread fungi) [47,48,49].

The available studies make it clear that there are several factors affecting the resistance to moisture (mold) [1,50,51,52,53,54,55], which are the following:Polymer type (Simonsen et al., 2004 showed higher protection of products against H_2_O via the application of the HDPE matrix, while the use of biopolymers was illustrated by Candelier et al., 2019);Modification of wood flour (Wei et al., 2013 used esterified wood flour from poplar, which increased the moisture resistance/dimensional stability when compared with a sample where the flour was not modified);Type of wood flour (Xu et al., 2015 compared six types of wood flour and their impact on resistance in different directions. For more information, see Figure 5);Additives and biocides (Verhey et al., 2001 and Klyosov, 2007 showed that the application of zinc borate, ZnB, increased the resistance to wood-decaying fungi. Similarly, the application of a 3% CCC regulator was performed in a study by Lu et al., 2008);Increasing the content of thermoplastics at the expense of wood flour.

It should be noted that moisture penetration into WPC materials is more than 20 times slower compared to conventional woody plants (the water absorption of a WPC is 0.3–1.9%, and the water absorption of pine wood is 17.2%) [1].

The WPC components themselves are flammable. The flash point of wood is 275 °C; for plastics, the temperature values are higher (e.g., PE = 360–367 °C). For wood-filled plastics, the flash point is around 400 °C, depending on the thermal conductivity, exact composition, amount of additives, and sample size. The monograph by the Russian author Klyosov, 2007 defined the flammability of materials based on the flame spread index (FSI), where individual materials are assigned indices; for example, the WPC profile from a PVC matrix and wood flour has a FSI of 25–60, the full profile from the HDPE matrix and wood flour has a FSI of 80–100, etc. Examples of the appearance of PS and WPC panels after burning are presented in Figure 6 [1].

The Kiaei study, 2017, pointed to the influence of zinc oxide with nanoparticles on the mechanical properties, flammability, and morphological characteristics of WPC samples. The bending strength of the WPC samples with zinc oxide as an additive was 79.9% higher than the WPC sample without an additive. Similarly, the additive had a positive effect in terms of flammability, where the time to ignition of the WPC increased by more than 100% and the burning rate decreased by 46% [56].

The exposure of WPC materials to ultraviolet radiation results in a loss of color on the surface of the material and slight changes in relation to the mechanical properties (more pronounced changes can be caused by a combination of the following undesirable factors including the oxidation of the surface of the material and changes in crystallinity and degradation caused by moisture). WPC profiles exposed to UV radiation (2000 h of exposure) became 13% lighter compared to the original shade, and the flexural modulus was reduced by 12% (Morrell et al., 2006); similar results were obtained by Stark and Matuana, 2006. In order to minimize the adverse effects of weather, additives are added to the mixture, including UV stabilizers, pigments, and antioxidants (protection against photodegradation, changes caused by heat, and color changes during outdoor applications) [57,58].

The properties of composite materials based on WPCs are impacted by individual component compositions. In general, it can be stated that increasing the ratio of filler or wood flour in relation to the reinforcement influences a decrease in flexural and tensile strength, an increase in modulus and elasticity, and a decrease in final product density. Negative impacts have increased the ratio of wood flour to water absorption (and thickness swelling) [24,59].

## 4. Conclusions

Since their initial commercial application in 1906, there has been a significant advancement in the composition and properties of composite materials with natural fibers. They no longer represent merely a “cheap” alternative but rather a fully-fledged material with specific properties. In the presented article, we reviewed and analyzed the compositions of materials with natural reinforcements and their mechanical/user properties. The main conclusions drawn from the available publications are as follows: The application of composite materials with natural reinforcements can contribute to reducing the carbon footprint (natural fibers can fully replace some types of synthetic fibers);In addition to standard polymers, biodegradable plastics based on PLA, PHA, PHB, and PHBV can also be applied in the case of NFC (WPC) matrices (the processing temperature is a limiting factor);The final properties of products depend on several attributes, including the type of matrix/reinforcement, the volume of filler or fibers (in the case of fibers, also their orientation), the particle morphology, the technological process of filler/fiber production, the surface adhesion of the examined materials, the interphase formation between the components, etc.;Adding modifiers (additives) affects a wide range of properties that are mechanical/user-related. The type and percentage of the applied modifier should be determined based on the conditions in which the product will be applied. The portfolio of additives offered/used in the NFC (WPC) manufacturing process is currently sufficiently broad (compatibilizers, lubricants, biocides, pigments, flame retardants, fillers, etc.). Some additives serve multiple functions; for example, pigments with UV stabilizer functions.

## Figures and Tables

**Figure 1 polymers-16-01556-f001:**
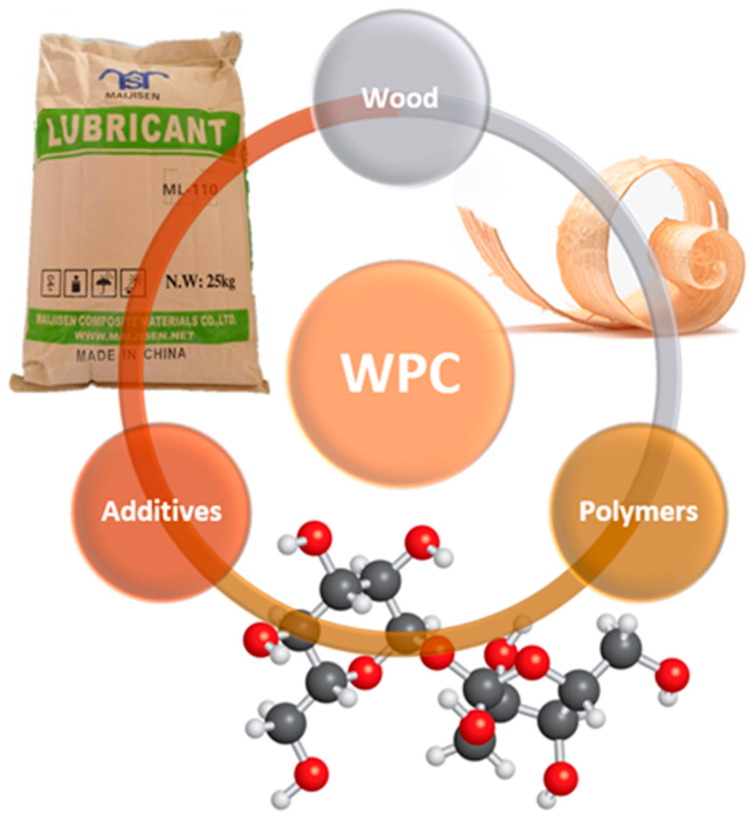
Composition of WPCs: wood reinforcement + polymeric matrix + additives.

**Figure 2 polymers-16-01556-f002:**
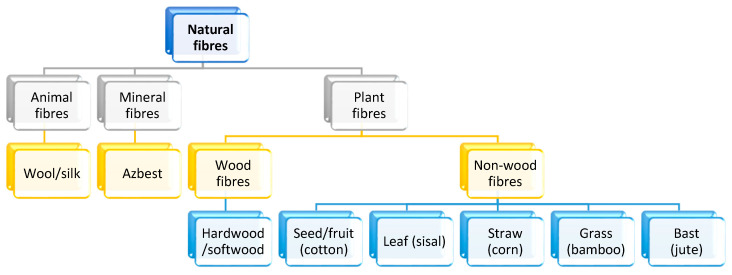
Classification of natural fibers and examples of non-wood fibers [12].

**Figure 3 polymers-16-01556-f003:**
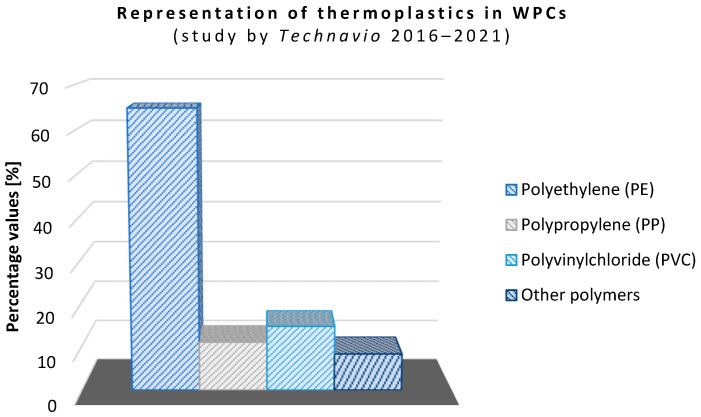
Representation of thermoplastics in WPC matrices (polyolefins and fluoroplastics) in a study by Technavio (2016–2021) [25].

**Figure 5 polymers-16-01556-f005:**
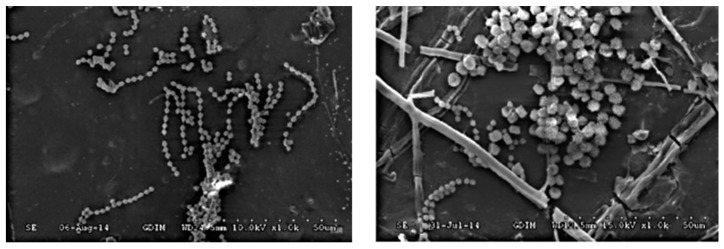
SEM images of WPC surfaces from different types of wood flour and emerging molds (type of wood flour: (**left**) side: *Eucalyptus grandis* × *E. urophylla*/(**right**) side: *Pinus massoniana*) [53].

**Figure 6 polymers-16-01556-f006:**
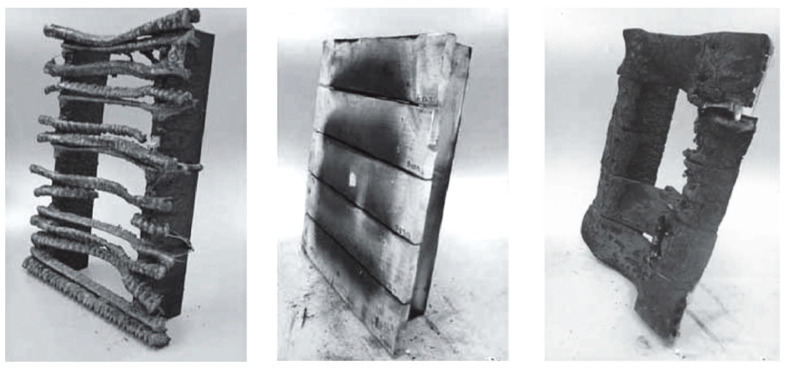
Panels from the PS (first from the left side) and WPC after ignition (composition of the second panel: 33% PE + 60% wood fibers + 7% minerals; composition of the third panel: 37% HDPE + 48% wood fibers + 15% minerals) [1].

**Table 2 polymers-16-01556-t002:** Recommended amounts of additives for WPCs with the PE matrix [2].

Material Function	Material	Percentage Volume
Matrix type	Polyethylene	Depending on other components
Reinforcement	Natural fibers	30–60%
Coupling agents	Maleinated polyolefin	2–5%
Lubricants	Stearates/esters/others	3–8%
UV stabilizers	HALS/benzophenone	0–1%
Fillers	Talc	0–10%
Pesticides	Zinc borate	0–2%
Dyes	Pigments (unspecified)	As required
Flame retardants	-	As required

**Table 3 polymers-16-01556-t003:** Modification of properties in relation to the applied additive [44].

WPC Sample Components	Bending Strength [MPa]	Flexural Modulus [MPa]
PPr—PIE	32.71	1929
PPr—PIE—PPgMA	44.34	2064
PPr—PIE—PPgIA	42.17	2271

## Data Availability

Not applicable.

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
