# Peer review of "A Concise Review of the Components and Properties of Wood–Plastic Composites"

_polymers, 2024, doi:10.3390/polym16111556_

Round 1
Reviewer 1 Report
Comments and Suggestions for Authors
This paper gives a concise review of wood-plastic composite: components and properties. It can help readers establish a general framework for understanding WPC materials and spark further research interest. Some minor comments:
1. The abstract can be better. In the abstract, the authors give too much space to describe the background, which should be given in the introduction part.
2. Please confirm the title of Figure 8
3. The quantitative influence of components on its properties should be reviewed.
4. It is suggested that the authors address reviewing of the mechanical properties of WPC members.
5. The conclusion should be carefully checked and rewritten according to the requirements of the journal.
Author Response
Thank you very much for your comments. For answear - please, see attachment.

Reviewer 2 Report
Comments and Suggestions for Authors
The submitted manuscript represents a significant contribution to understanding the Wood-Plastic Composite (WPC) and will, therefore, be of interest to the journal's readers.
I have only a few minor recommendations for the esteemed authors:
1) It would be good to give some main results from the review in the abstract.
2) At the end of the Introduction, it would be good to provide a historical overview of the WPC and address the need for the presented manuscript. That is, which issues are controversial and what the present review contributes to their clarification.
3) I would like to analyze the advantages and disadvantages of different fillers, matrices, and additives at the end of the Components of Composite Materials section based on the WPC section.
4) Please increase the quality (resolution) of Figure 3. (lines 178-179); The same recommendation applies to Figure 4 (lines 239-240).
The references cited are appropriate.
Author Response

(The authors gave the same response as above.)
